# Proteomic Profiling of COVID-19 Patients Sera: Differential Expression with Varying Disease Stage and Potential Biomarkers

**DOI:** 10.3390/diagnostics14222533

**Published:** 2024-11-13

**Authors:** Iman Dandachi, Ayodele Alaiya, Zakia Shinwari, Basma Abbas, Alaa Karkashan, Ahod Al-Amari, Waleed Aljabr

**Affiliations:** 1Research Center, King Fahad Medical City, Riyadh 11525, Saudi Arabia; iman.j.d@hotmail.com; 2Cell Therapy & Immunobiology Department, King Faisal Specialist Hospital and Research Centre, Riyadh 12713, Saudi Arabia; aalaiya@kfshrc.edu.sa (A.A.); szakia@kfshrc.edu.sa (Z.S.); 3Department of Biological Sciences, College of Science, University of Jeddah, Jeddah 21959, Saudi Arabia; bma_1403@hotmail.com (B.A.); askarkashan@uj.edu.sa (A.K.); 4Department of Basic Medical Sciences, College of Medicine, Dar Al-Uloom University, Riyadh 13314, Saudi Arabia; ahod.k.a@gmail.com; 5Public Health Authority, Riyadh 13352, Saudi Arabia; 6Institute of Infection, Veterinary and Ecological Sciences, University of Liverpool, Liverpool L3 5RF, UK

**Keywords:** proteomics, COVID-19, disease stage, biomarker, sera

## Abstract

**Background/Objectives**: SARS-CoV-2 is one of the viruses that caused worldwide health issues. This effect is mainly due to the wide range of disease prognoses it can cause. The aim of this study is to determine protein profiles that can be used as potential biomarkers for patients’ stratification, as well as potential targets for drug development. **Methods**: Eighty peripheral blood samples were collected from heathy as well as SARS-CoV-2 patients admitted at a major tertiary care center in Riyadh, Saudi Arabia. A label-free quantitative mass spectrometry-based proteomic analysis was conducted on the extracted sera. Protein–protein interactions and functional annotations of identified proteins were performed using the STRING. **Results**: In total, two-hundred-eighty-eight proteins were dysregulated among all four categories. Dysregulated proteins were mainly involved in the network map of SARS-CoV-2, immune responses, complement activation, and lipid transport. Compared to healthy subjects, the most common upregulated protein in all three categories were CRP, LGALS3BP, SAA2, as well as others involved in SARS-CoV-2 pathways such as ZAP70 and IGLL1. Notably, we found fifteen proteins that significantly discriminate between healthy/recovered subjects and moderate/under medication patients, among which are the SERPINA7, HSPD1 and TTC41P proteins. These proteins were also significantly downregulated in under medication versus moderate patients. **Conclusions**: Our results emphasize the possible association of specific proteins with the SARS-CoV-2 pathogenesis and their potential use as disease biomarkers and drug targets. Our study also gave insights about specific proteins that are likely increased upon infection but are likely restored post recovery.

## 1. Introduction

Coronaviruses (Cov) are a large family of respiratory viruses that have caused worldwide health concerns in humans as well as in animals. Viruses in this family are characterized by a crownlike shape, with a genome composed of a non-segmented positive sense, single-stranded RNA that ranges from 27 to 32 kilobase-pairs in size [1]. The coronavirinae family encompasses four genera: alpha, beta, gamma, and delta [2]. Among these, alpha and beta have been considered major concerns due to their ability to cross the animal barrier and infect humans [3]. So far, seven coronaviruses caused human diseases and included the alpha-cov NL63 and 229E, and the beta-cov: OC43, HKU1, SARS-CoV, MERS-COV, and the recently emerged SARS-CoV-2 virus [4]. Among these seven, the SARS-CoV, MERS-COV, and the SARS-CoV-2 have been regarded as highly pathogenic to humans [5]. This perception is due to their ability to cause severe infection that can progress to multi-organ failure and death. Following its first detection in 2019, SARS-CoV-2 is still circulating and evolving worldwide with several variants of concerns, of interest, and under monitoring being surveilled [6].

The clinical manifestation of SARS-CoV-2 is highly variable and ranges from an asymptomatic infection to pauci-symptomatic to severe disease. The severe form of the disease is manifested by a range of systemic symptoms that varies from fever, cough, and chills to fatigue and shortness of breath. The severity and complications of the infection relies on the possibility of developing pneumonia, as well as heart, liver, or respiratory failure [1]. Patients might also risk the development of acute respiratory distress (ARDS) and multi-organ failure [7,8]. In this context, it is essential to provide novel tools that assist in identifying patients at a higher risk of severe COVID-19 and accordingly set appropriate patient based clinical management and therapeutic intervention [9,10,11,12]. For this, recent studies have suggested “blood proteomics” as an effective tool to explore differentially expressed proteins that could be targeted in infected patients with different disease stage and/or severity. Indeed, several studies have found that interleukin-6, lactate dehydrogenase, serum amyloid A, as well as C-reactive proteins and procalcitonin are predictors of COVID-19 severity [13,14]. Proteins are key effectors in most cellular functions. Analyzing dysregulated proteins in infected patients will unveil the occurring system-perturbation, guiding thus our understanding of the host-pathogen interaction. This action will also strengthen our defenses for future pandemics [11]. In Saudi Arabia, proteomic studies comparing healthy subjects to infected and recovered patients are scarce. The aim of this work is to determine the proteomic profiles of moderate, under medication, and recovered SARS-CoV-2 patients compared to healthy subjects. Our attempt is to strengthen the knowledge on possible biomarker predictors of the disease stage in COVID-19 patients.

## 2. Materials and Methods

### 2.1. Samples and Data Collection

Eighty blood samples were collected from patients admitted at the Prince Mohammed Bin Abdulaziz Hospital (PMAH), Riyadh, KSA, diagnostic-laboratory during 2020 and suspected for COVID-19 disease. These samples included 20 patients (>15 years old) from each of the following categories: healthy, moderate disease, under medication, and recovered. All included subjects did not receive the SARS-CoV-2 vaccine. Patients were characterized as healthy individuals if they had no history of respiratory illnesses, nor smoking, or obesity, characterized in a moderate stage of infection, if they were hospitalized but did not start treatment yet (samples collected between 4 and 9 days [average of 6.5 days] from the day of reporting positive COVID-19), patients under medication if they received medical treatments upon admission to the hospital (samples collected between 10 and 14 days [average of 12 days] from the day of reporting positive COVID-19), and recovered patients if the samples were collected between 14 and 20 days [average of 17 days] from the day of reporting a positive COVID-19 test. Demographic data, i.e., age, gender, and nationality were collected for all included subjects. 

### 2.2. Serum Preparation and Protein Extraction

We took adequate precautions in working with biomaterials; thus, all samples were heat inactivated at 65 °C for 30 min and in compliance with BSC IIA protocols prior to processing for a proteome analysis as previously reported [15]. Prior to a liquid chromatography mass spectrometry analysis, serum samples were depleted of common high-abundance serum proteins using a human albumin removal kit from Agilent Technologies (Santa Clara, CA, USA). Extracted protein was then quantified using Qubit™ Protein and Protein Broad Range (BR) Assay Kits. Owing to low throughput of an LC-MS- proteomic analysis, 20 samples in each cohort category were pooled together and run for the assessment of proteomic profiles. Each sample composition of an analysis group samples was subjected to multiple LC/MS runs (3 to 4 times), and the average of multiple runs was evaluated for statistical analysis.

### 2.3. Protein Characterization by Label-Free Liquid Chromatography/Mass Spectrometry

The protein concentrations of all samples were normalized, and a total of 100 μg representing the protein contributed by each sample composition of an analysis cohort pooled samples was subjected to an in-solution tryptic digested prior to the LC–MS/MS analysis as previously described [15]. Briefly, depleted serum protein fractions were heat-denatured at 80 °C for 15 min, followed by a reduction in 10 mM of DTT at 60 °C for 25 min, then alkylated in 10 mM of iodoacetamide (IAA) for 30 min, and slowly agitated at room temperature in the dark. The enzyme trypsin (Promega, US) at a concentration of 1 μg/μL was added (50:1 sample/trypsin ratio) for overnight digestion at 37 °C. All the samples were spiked with yeast alcohol dehydrogenase (ADH; P00330) as an internal standard for absolute quantitation. Equal amounts of digested peptides and 3 μg of each sample cohort was injected for the LC–MS/MS analysis using one-dimensional Nano Acquity liquid chromatography coupled to Synapt G2 HDMS on a Trizaic Nano-flow source (Waters, Manchester, UK). The instrument calibration and optimization settings were conducted using the MassLynx tune page prior to analysis. The MS data were acquired in an *m*/*z* range of 50–2000 Da with a gradient acquisition run time of 120 min using data-independent acquisition (DIA)/ion-mobility separation experiments (HDMSEs), operated in resolution and positive polarity modes as previously described [15]. Each sample was analyzed in triplicate runs using the Mass Lynx platform (Version. 4·1, SCN833) (Waters, Manchester, UK).

### 2.4. Bioinformatic Analysis

The acquired raw MS data were processed, and database searching was accomplished using the Progenesis LC-MS proteomic data analysis software (Progenesis QIfP V4.0 (Waters, Manchester/Nonlinear, Newcastle, UK) as previously described [15]. The acquired list of peptide ions was queried using the non-redundant UniProt/SwissProt human-specific (Homo sapiens) protein sequence database for protein identification (www.uniprot.org) [16]. The generated data were filtered using a multivariate statistical analysis and differential expression analysis. An algorithm in the licensed-based Progenesis QI for proteomics ((Nonlinear Dynamics, Newcastle/Waters, UK) was used for the data analysis. We used the default method of normalization to all proteins based on ratiometric data in log space in the program Progenesis QI for proteomics. The program used all compound ion abundances and generated a multiplication factor to give a normalized abundance based on a known spiked protein abundance. We made measures to overcome the associated problems of multiple testing and the False Discovery Rate (FDR) and applied the adjusted *p*-value or q-value calculated by the imbedded algorithm in the licensed-Progenesis QIfP program. We applied multivariate data analysis to identify only statistically significant regulated proteins of normalized label-free quantifications of protein abundance. Furthermore, we performed ‘Hi3′ absolute quantification using a known protein as the internal standard (ADH, P00330) for the absolute amount of each identified protein. The identified proteins were further filtered and limited to only those with confidence identifications with ≥2-identified peptides, with significantly altered proteins (*p* ≤ 0.05) (ANOVA), and a fold change of more than 2. The Synapt G2 HDMS that was used in this study was very reliable in capturing molecules within the detection limit of approximately 25 fmol. Therefore, the identified proteins that fulfilled the criteria were considered reliable. Heatmap, volcano, and horizontal bar plots were conducted using SRplot [17]. A functional protein analysis was performed using the Retrieval of Interacting Gene/Protein database (STRING v.12.0) [18] and cytoscape software v3.10.1 [19] (Figure 1). 

## 3. Results

### 3.1. General Characteristics of the Studied Population

The patients’ demographic characteristics are summarized in Table 1. The majority of patients were non-Saudis (62.5%); non-Saudi nationalities included Yemeni, Syrian, Egyptian, Pakistani, Sudanese, and others. Ages ranged from twenty-one to ninety-one years old. Furthermore, the distribution of the female gender ranged from 30 to 40%, while the male gender distribution ranged from 60 to 70%, in all four categories.

### 3.2. Proteomic Profiles per Disease Category

Owing to the low throughput of LC-MS proteomic analysis, 20 samples in each cohort category were pooled together and run for the assessment of proteomic profiles. We recognized the inherent limitation of the analysis of pooled samples in any analysis platforms including the MS-based analysis with potential risks of confounding the results if proper care was not as widely reportedly. However, in this study, we have taken steps to mitigate potential artifacts arising from pooling samples. We always undertake an initial screening step in maintaining homogeneity within an analysis group prior to pooling. 

Four-hundred thirty-nine proteins were identified in all four categories. Following the application of filters (peptide count > 1, max fold change ≥ 2 and ANOVA *p* < 0.05), two-hundred eighty-eight were found to be differentially expressed significantly (Appendix A). An unsupervised hierarchical cluster analysis showed a formation of four distinct clusters of analyzed categories of patients (Figure 2). The hierarchical tree formed two main branches: one containing the healthy subjects and one containing moderate, under medication, and recovered COVID-19 patients.

### 3.3. Differential Expression of Proteins per Disease Category Compared to Healthy Subjects

Overall, one hundred sixty-one proteins were upregulated in at least one category, whereas one hundred forty-one proteins were downregulated in at least one category, both compared to healthy patients. Of the up-regulated proteins, seventy-eight were upregulated in all moderate, under medication, and recovered patients, with the most being MASP1, MYL5, CCDC39, LYAR, IGHV3-53, KCTD4, and DCUN1D5, in addition to others such as ITIH3, GSN, and ACTB. On the other hand, sixty-six were downregulated in all three categories, with the most being IGHM, APOM, C1QC, APOC4, GREB1L, SHBG, IGHV5-51, FGA, IGHV3-15, SGSM3, and FBLN1 (Figure 3). Moreover, fifty-five were up-regulated in moderate and/or under medication patients, with the most being SERPINA7, CNGA4, E3 15.3 kDa, KIF16B, PON3, and LCAT. On the other hand, fifty-one were downregulated in the same categories with the most being IGKV4-1, IGHV1-69D, IGKV3-15, and DEFA1.

### 3.4. Functional Analysis of Dysregulated Proteins Compared to Healthy Subjects

Among the dysregulated proteins both in all three categories compared to the healthy subjects, the most common enriched biological pathways were biological regulation, response to stress and stimulus, as well as immune system responses (Figure 4).

Notably, when looking at the Wiki-pathways (22) (23), we found that the most enriched ones belong to the “network map of SARS-CoV-2 signaling pathways” which contains, notably, the upregulated proteins CRP, LGALS3BP, ZAP70, IGLL1, and SAA2, the “fatty acids and lipoproteins transport in hepatocytes” which encompasses the upregulated proteins LGALS3BP, MIB1, TFRC, SAA2, and the “complement and coagulation cascades, complement system”, which include the upregulated CPB2, CRP, C9, C6, KLKB1, MASP1, and MBL2 proteins. Moreover, the “pathogenesis of SARS-CoV-2 mediated by nsp9-nsp10 complex”, which includes the three upregulated proteins CRP, ZAP70, and IGLL1, was also found (Figure 5). 

For proteins that were dysregulated only in moderate and/or under medication patients compared to healthy subjects, the Wiki-pathways found were the “network map of SARS-CoV-2 signaling pathways”, which include the upregulated proteins SERPINA10, LRG1, PF4, CPN1, APOC1, and the “RAS and bradykinin pathways in COVID-19”, which include the three upregulated proteins AGT, CPN1, PRKG1, and the “statin inhibition of cholesterol production” which also includes the upregulated proteins APOE, LCAT, and APOC1 (Figure 6).

### 3.5. Differential Expression of Proteins in Moderate/Under Medication Patients Compared to Recovered

In total, one hundred thirty-eight proteins were upregulated in moderate and under medication patients compared to recovered. Of these seventy-two that were upregulated in both moderate and under medication, thirty-eight were in moderate only and twenty-eight in under medication only. The most common among the seventy-two, significantly, upregulated proteins were SERPINB1, RBSN, GAG-POL, TBC1D1, PIK3C2B, MAN1A2, and CFHR2. On the other hand, one-hundred-thirteen proteins were downregulated in moderate and under medication patients compared to recovered. Of these, fifty-five were downregulated in both categories, thirty in moderate only, and twenty-eight in under medication only. The most common proteins significantly downregulated in both categories together, were IGLV1-47, IGHV3-7, IGHV1-3, IGHV4-4, GPD1L, IGLV1-51, and Ig α-2 (Figure 7). Among those upregulated in moderate only, the most common were CFHR1, SERPINA1, SAA1, and CLSPN, whereas the proteins that were mostly downregulated were CD5L, IGHM, IFT172, and IGFBP3. For under medication patients, the most common upregulated were IGFBP3, and CD5L, whereas the most common downregulated proteins were IGHV6-1 and IGLV2-18.

### 3.6. Functional Analysis of Dysregulated Proteins in Moderate/Under Medication Patients Compared to Recovered

Enriched Wiki-pathways in dysregulated proteins together in moderate, as well as in under medication patients compared to recovered equally included the “network map of SARS-CoV-2 signaling pathway”, “complement system”, and “complement and coagulation cascades”. These were all composed of upregulated proteins (except for the C1QA that was downregulated), CFI, C8A, FGA, KLKB1, MASP1, and C2 (Figure 8). 

### 3.7. ROC Curve and Potential Biomarker Identification

In this analysis, proteins with a non-significant difference between healthy and recovered subjects and that were also either (a) upregulated significantly in moderate and under medication patients compared to healthy and recovered, or (b) downregulated significantly in moderate and under medication patients compared to healthy and recovered, were taken and assessed using the ROC curve test. We found that the upregulated SERPINA7, SERPINA10, PDHB, CPO, CNDP1, DNA2, THSD4, LCAT, STX1A, HSPD1, TTC41P, AGT, GLOD4, Env and E3 15.3 kDa are discriminators between healthy/recovered and moderate/under medication subjects, with an AUC score of 1 for all 15 proteins except for GLOD4, which had an AUC score of 0.974 (Figure 9). Interestingly, using cytoscape software v3.10.1 (21), AGT was involved in “SARS-CoV-2 and ACE2 receptor: molecular mechanisms”, “SARS-CoV fibrosis”, and “RAS and bradykinin pathways in COVID-19” curated pathways. In the network map of SARS-CoV-2 signaling pathway, AGT could be involved in “lung injury” in plasma, while SERPINA10 could be involved in severe/critical patients in peripheral blood mononuclear cells [20,21]. Moreover, as for downregulated genes, IGKV3-15, ADH1, MLF2, DEFA1, IMMMUNO, IGKV4-1, TEX12, DDX56, and RXRA were also discriminators between these two categories with all having an AUC score of 1 (Appendix A).

### 3.8. Differentially Expressed Proteins Between Under Medication and Moderate Patients

Out of two-hundred eighty-eight differentially expressed proteins between all four categories, one-hundred fifty-six were significantly differentially expressed between moderate and under medication patients: seventy-three upregulated and eighty-four downregulated. Compared to moderates, PON3, IGFBP3, CD5L and IGKV1-17 were the most significantly upregulated proteins in under medication patients. On the other hand, FAM167A, SERPINA7, IGHG2, HP, and TGFBI were the most significantly downregulated. String PPI and cytoscape software analysis revealed that six proteins were involved in the “network map of SARS-CoV-2” which included the down APOC1, APOL1, APOM, CFP, GSN and IGFBP3 proteins. APOC1 and GSN are markers of disease in severe/critical patients. On the other hand, within the same network of SARS-CoV-2 map, fourteen proteins were downregulated including ALB, CPN1, CRP, HP, IFIHI, IGHG2, ITIH3, ITIH4, LBP, LGALS3BP, LRG1, MIB1, PF4, SAA1, SAA2, and ZAP70 (Figure 10). These proteins might be involved in lung injury, neutrophil activation, T cell receptor and immunoglobulin subunits, as well as TCR signaling kinases. Furthermore, the downregulated proteins SERPINA7, HSPD1, and TTC41P were among the key potential biomarkers that are, as per the ROC curve performed in our study, key discriminators between moderate/under medication patients and healthy/recovered subjects.

## 4. Discussion

In recent years, it has been suggested that perturbation of the plasma protein abundances, upon infection, is a good marker of the pathophysiological changes that the body is undergoing [22]. SARS-CoV-2 is one of the viruses that had worldwide impact, and exploring its mechanistic way in different populations is of vital importance. Several blood biomarkers such as platelet distribution width [23], coagulation biomarkers such as prothrombin and fibrinogen, as well as chemokines and anti-inflammatory cytokines have been correlated with COVID-19 severity in infected patients [24]. Most of the studies in the literature explored differentially expressed proteins in patients with different disease severities [22]; however, in our study, we have compared healthy subjects to hospitalized patients who started or did not start yet treatment and to those who recovered from the disease. We attempted to find potential biomarkers that are increased upon infection compared to healthy subjects but are also decreased upon recovery. Using label-free mass spectrometry, we were able to identify two-hundred eighty-eight proteins that were differentially expressed among the four categories. Dysregulated proteins were mainly associated with biological regulation, response to stimulus, and immune system process. This finding is in accordance with other studies in the literature that found that the most common perturbed pathways are related to immunological responses [25,26]. Specifically, fourteen proteins are displayed with at least a 200 max fold change, including immunoglobulins, ASB3, and ZAP70. Interestingly, the Ankyrin repeat and SOCS box protein 3 (ASB3) that has a max fold change of 493 in moderate patients compared to recovered in our study, was found in a study conducted by Cheng et al., to be upregulated, in the presence of RNA viruses. According to their findings, ASB3 controls the antiviral signaling activation by acting as a negative regulator [27]. ZAP70, which is tyrosine-protein kinase, has been known to play a role in T-cell-receptor (TCR) signaling in peripheral T cells, as well as thymocytes [28]. This protein is also involved in the “Pathogenesis of SARS-CoV-2 mediated by nsp9-nsp10 complex” pathway according to the Cytoscape software [19]. Moreover, compared to our findings, other studies in the literature, exploring proteomic profiles of COVID-19 patients compared to healthy subjects, found dysregulated proteins related to other pathways, including the ones related to platelet degranulation and macrophage function [12].

We found that, compared to healthy subjects, dysregulated proteins were specifically mainly part of the SARS-CoV-2 network map, complement activation, and coagulation cascades as well as the statin inhibition of cholesterol production. Interestingly, we found upregulated proteins that are, as per the network map of SARS-CoV-2 related to lung injury “C-reactive protein (CRP), and angiotensin (AGT)” and those that could be considered as markers in severe/critical patients: the upregulated CRP, complement factor I (CFI), Inter-alpha-trypsin inhibitor heavy chain H3 (ITIH3), the Galectin-3-binding protein (LGALS3BP), serum amyloid 2 (SAA2), the protein Z-dependent protease inhibitor (SERPINA10), and the downregulated gelsolin (GSN) and Apolipoprotein C-I (APOC1). These proteins have been associated with disease severity in COVID-19 patients in several previously published studies [9,11,12,22,25,29,30]. CRP is a well-known marker of inflammation and a predictor of disease worsening in SARS-CoV-2 patients [31,32]. Indeed, in one study, it was found that significant elevation in the abundance of CRP in hospitalized COVID-19 patients is associated with QTc interval prolongation, increasing thus the risk of ventricular arrythmias and sudden death [33]. The increase in CRP abundance is likely the result of macrophage activation, upon the binding of SARS-CoV-2 to the ACE receptor and the subsequent activation of the interleukin-6 (IL-6) response [34]. Interleukin-6 is among the first key drivers of the host cell response towards the SARS-CoV-2 infection and is responsible for the eliciting and regulation of vital inflammatory proteins, including the serum amyloids SAA1/2 and ITIH ones [11,35], which were also found to be upregulated in our study, compared to healthy patients. Indeed, it has been suggested that, with other proteins such as Azurocidin 1, ITIH4 could help in avoiding the SARS-CoV-2 complete elimination, in the bodies of long-term persistent COVID-19 patients [36]. Moreover, when compared to healthy subjects, ITIH4, together with LGALS3BP, has been shown to be elevated in all SARS-CoV-2 infected patients, regardless of disease severity [29]. LGALS3BP is another biomarker that is virus-induced and can lead to cytokine storms in infected patients due to the subsequent elevation of pro-inflammatory cytokines such as IL-6 and the tumor necrosis factor alpha [37,38]. In one study, conducted by Gutmann et al., LGALS3BP was found to be significantly increased in ICU patients compared to uninfected individuals [39]. In our study, on the contrary of other identified biomarkers, we have found that GSN was downregulated in all COVID-19 patients compared to healthy individuals. This finding is in contrast of a study conducted by Pagani et al. who reported an increased plasma abundance of GSN in mild COVID-19 patients, suggesting its protective effect from the exacerbation of the disease, in moderately infected individuals [25]. Moreover, in our study, we found that APOM and APOH were downregulated in all three categories compared to healthy subjects. This finding is in accordance with a study conducted by Shen et al., who found that, compared to heathy as well as non-SARS-CoV-2 infected patients, APOM was downregulated in severely infected COVID-19 subjects [13]. It is worth mentioning that the dysregulation of the aforementioned proteins in all moderate, under medication patients as well as those that are recovered, might explain the longtime persistence of COVID-19 symptoms. For instance, it has been observed that perturbation of the plasma proteins can last up to six weeks following the first SARS-CoV-2 positive test result [40]. It should also be noted that the differential expression of proteins among different categories, in different studies, could arise from the technique used, definition of the disease category, in addition to the time point of sample collection from disease onset. 

Interestingly, we found in our study, fifteen proteins including angiotensin (AGT) and Protein Z-dependent protease inhibitor (SERPINA10), with no significant difference between healthy and recovered individuals but were significantly upregulated in both in moderate and under medication patients with an AUC score of 1. Both AGT and SERPINA10 could be related to lung injury and severity in SARS-CoV-2 patients, respectively. According to the cytoscape software, AGT is part of the RAS and bradykinin pathway in COVID-19, and in SARS-CoV fibrosis as well as lung injury in infected patients. In the RAS and bradykinin pathway, AGT is cleaved to give angiotensin I, which is also cleaved to give angiotensin II. Angiotensin II then induces several reactions, including vasoconstriction, oxidative stress, pro-apoptosis, pro-fibrosis, as well as pro-inflammatory signalizations [41]. These latter can then result in hypercoagulation, cytokine storms, acute respiratory distress syndrome and accordingly multiple organ damage [42]. Indeed, studies have shown that the levels of angiotensin II are increased in COVID-19 patients and that plasma levels correlates with the degree of lung injury [43]. 

Interestingly, one of the proteins that was upregulated in moderate/under medication patients compared to healthy/recovered was the Lecithin cholesterol acyltransferase (LCAT). This protein is part of the “statin inhibition of cholesterol production”, that was in our study mediated by upregulated proteins (including the apolipoprotein C-I, APOC1) in COVID-19 patients. LCAT is an enzyme that promotes the transport of extra cholesterol from blood and tissues to the liver for excretion [44]. As for APOC1, several studies have found that the latter is reduced in severe/critical SARS-CoV-2 patients [45,46]. Apolipoprotein also mediates the transfer of cholesterol from peripheral tissues to the liver, promoting thus anti-inflammatory, anti-infectious effect [45]. For instance, it has been suggested that cholesterol supports viral infection; the decrease in the proteins mediating its transport in severe patients and their increase in those with moderate disease [46] emphasizes on their possible role in COVID-19 severity, as well as their possible use as drugs that target cholesterol metabolism and transport in infected patients. Indeed, studies have shown that statin treatment is associated with a better disease prognosis and reduction in death risk in COVID-19 patients [47,48,49].

It is worth mentioning that, interestingly, when patients who took medication were compared to those who did not start medication yet, a significant decrease in proteins involved in lung injury, neutrophil activation and immune responses, was observed; but also, in the SERPINA7, HSPD1, and TTC41P, the proteins that were found to be possibly discriminatory between moderate/under medication patients and healthy/recovered subjects. This finding emphasizes their possible role in the pathogenesis of the COVID-19 disease. Indeed, in severe patients, the mitochondrial 60 kDa heat shock protein (HSPD1) has been previously reported to be a biomarker of cardiac malfunction [50]. On the other hand, Survana et al. have also found that SERPINs plasma proteins, including SERPINA7 are a possible target of drug therapy in COVID-19 patients [50]. 

Our study has several limitations. The first one is the incomplete clinical characteristics of included subjects (such as SARS-CoV-2 variant, comorbidities) and age semi-matching between controls and infected groups (Figure 1). The presence of pre-existing comorbidities as well as the age of SARS-CoV-2 patients are factors that can influence the COVID-19 prognosis. Another limitation is the small sample size in each category of subjects. Indeed, at the beginning of the pandemic and during the lockdown phase, the collection of samples was challenging in the country, hindering the collection of a larger sample size and the full control of possible confounders in the studied population. 

## 5. Conclusions

Nevertheless, taken together, our study gave insights on of the proteomic profiles of infected COVID-19 patients compared to healthy and recovered individuals in the Saudi population. New potential biomarkers that might be associated with SARS-CoV-2 infection, such as HSPD1 and the tetratricopeptide Repeat Domain 41 (TTC41P) were also found. Future studies should aim at exploring the possibility of using these proteins as potential targets for drug development against the SARS-CoV-2 virus.

## Figures and Tables

**Figure 1 diagnostics-14-02533-f001:**
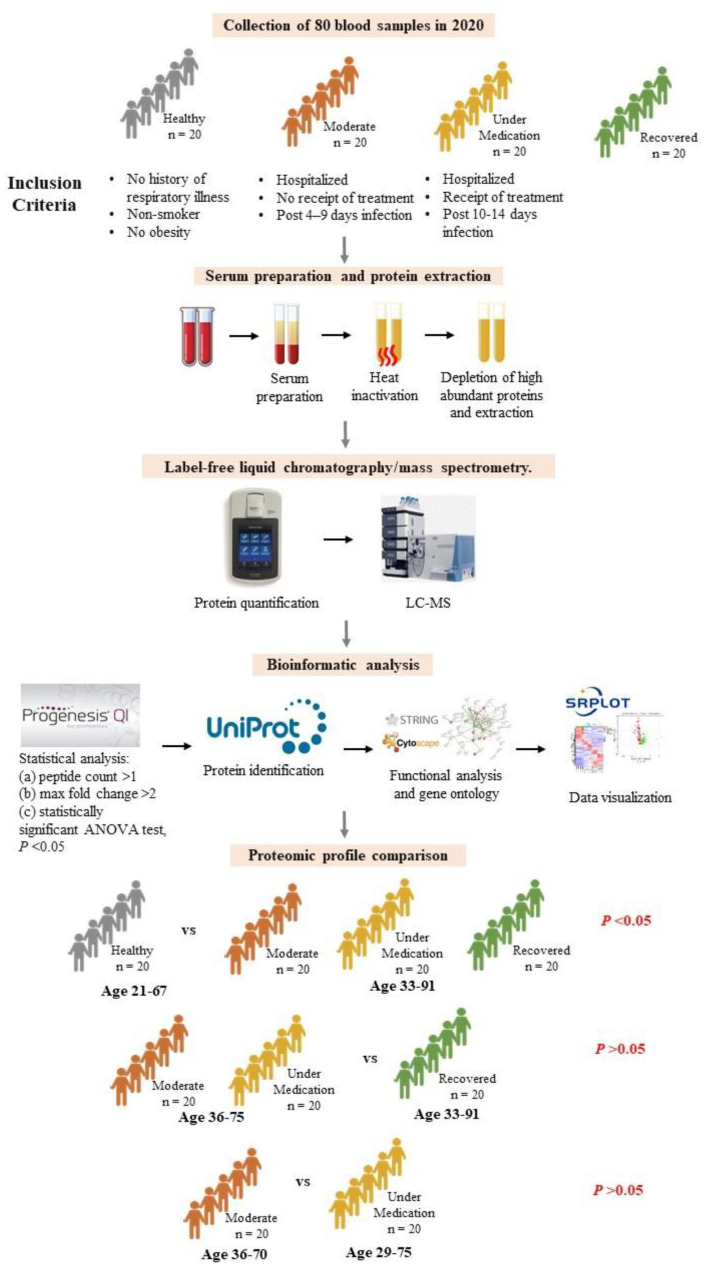
Flow chart showing the workflow of this study. *p* value showing significance or non-significance difference between ages of different categories compared.

**Figure 2 diagnostics-14-02533-f002:**
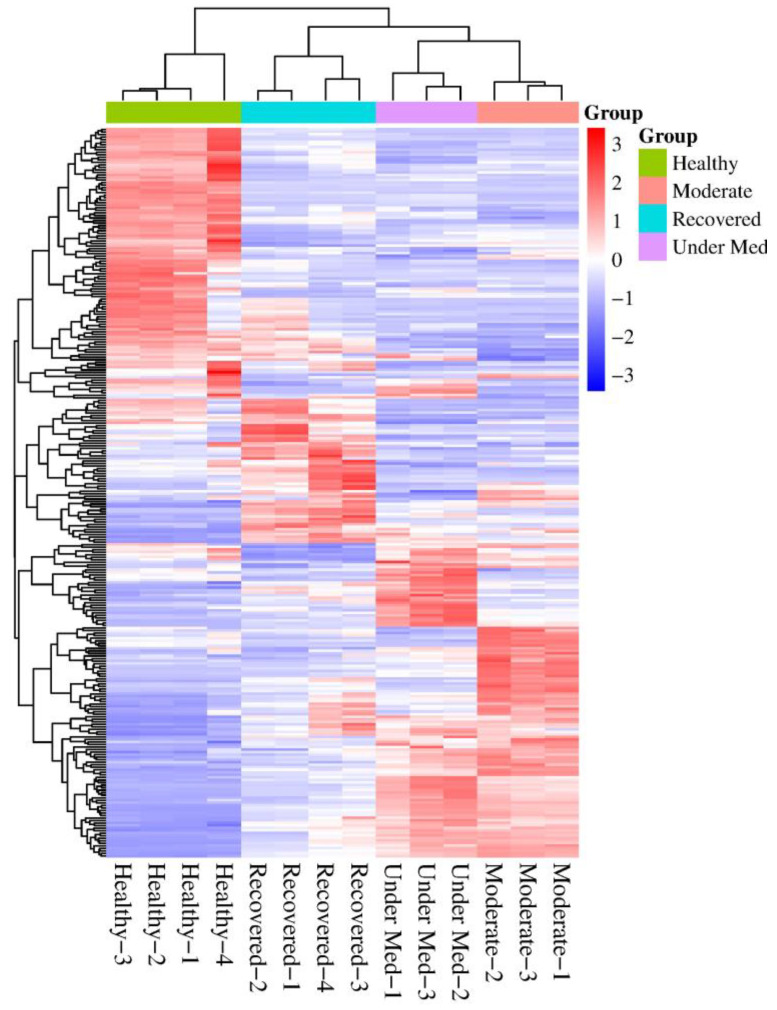
Hierarchical clustering of healthy, moderate, under medication (under Med) and recovered patients based on the differential expression of obtained proteins.

**Figure 3 diagnostics-14-02533-f003:**
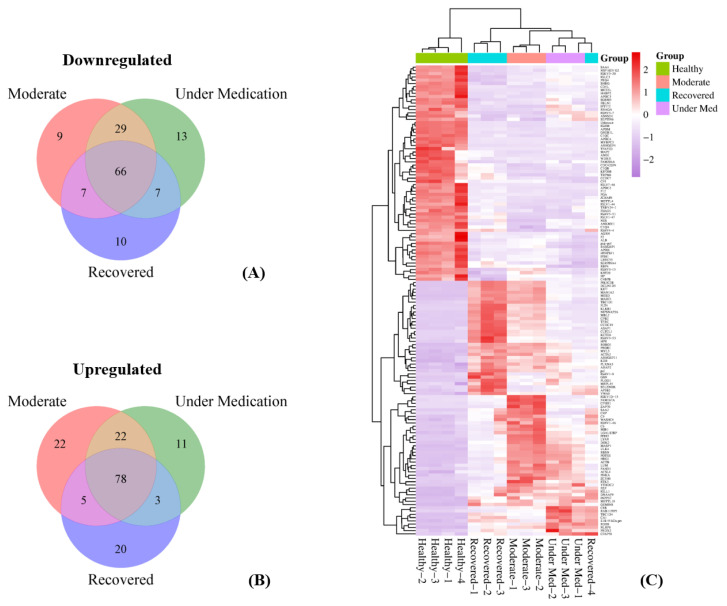
Upregulated and downregulated proteins in patients versus healthy subjects: (**A**) vein representation of upregulated proteins in all four categories, (**B**) vein representation of downregulated proteins in all four categories, and (**C**) hierarchical clustering of upregulated and downregulated proteins in all three categories compared to healthy subjects.

**Figure 4 diagnostics-14-02533-f004:**
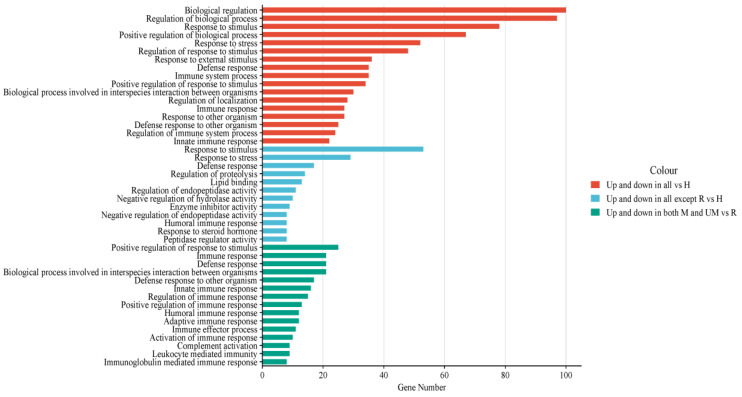
Enriched go pathways of dysregulated protein in different comparisons. H = healthy, M = moderate, UM = under medication, and R = recovered.

**Figure 5 diagnostics-14-02533-f005:**
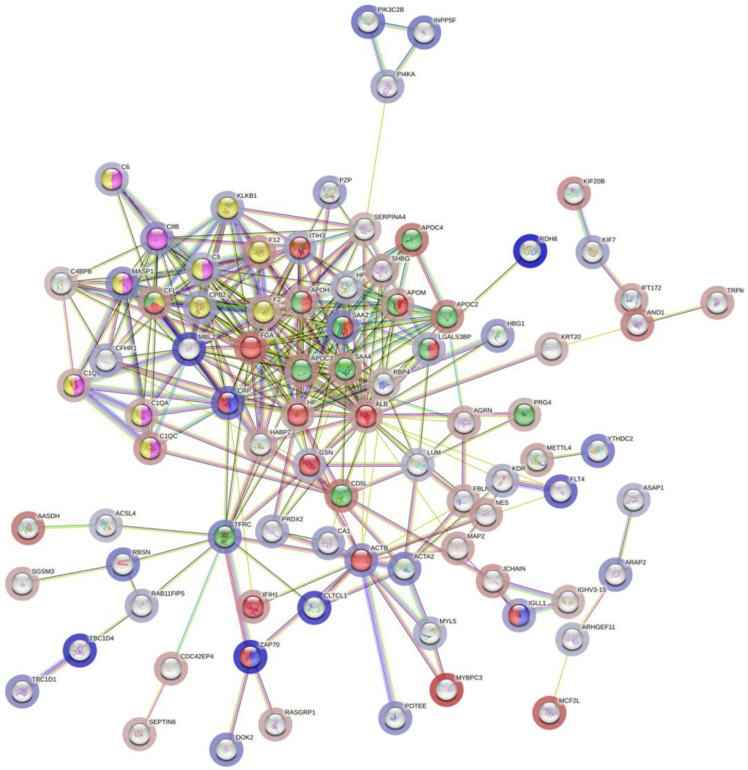
STRING PPI showing the protein interaction of common upregulated and downregulated proteins in all categories compared to healthy subjects. Edges represent protein–protein associations. Purple and light blue edges represent known interactions from experiments and curated databases, respectively. Red bubbles represent the network map of SARS-CoV-2 signaling pathways. Green represents the fatty acids and lipoprotein transport in hepatocytes. Yellow represents complement and coagulation cascades. Pink the complement activation pathway and blue the pathogenesis of SARS-CoV-2 mediated by nsp9-nsp10 complex.

**Figure 6 diagnostics-14-02533-f006:**
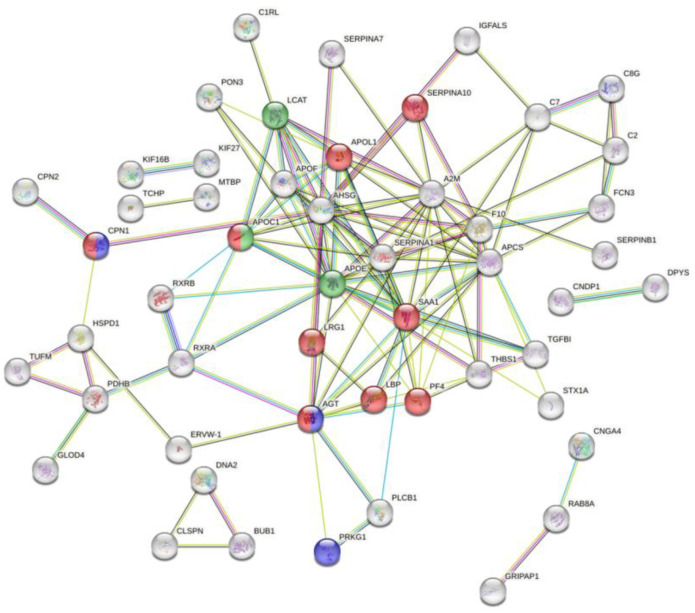
STRING PPI showing the protein interaction of common upregulated and downregulated proteins in moderate and/or under medication and not recovered compared to healthy subjects. Edges represent protein–protein associations. Purple and light blue edges represent known interactions from experiments and curated databases, respectively. Red bubbles represent the network map of SARS-CoV-2 signaling pathways. Blue represents the RAS and bradykinin pathways in COVID-19. Green represents the statin inhibition of cholesterol production.

**Figure 7 diagnostics-14-02533-f007:**
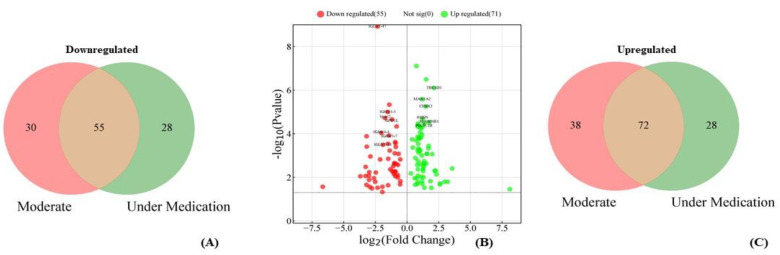
Upregulated and downregulated proteins in moderate and under medication versus recovered patients: (**A**) Venn representation of upregulated proteins, (**B**) volcano plot of upregulated and downregulated proteins, and (**C**) Venn representation of downregulated proteins.

**Figure 8 diagnostics-14-02533-f008:**
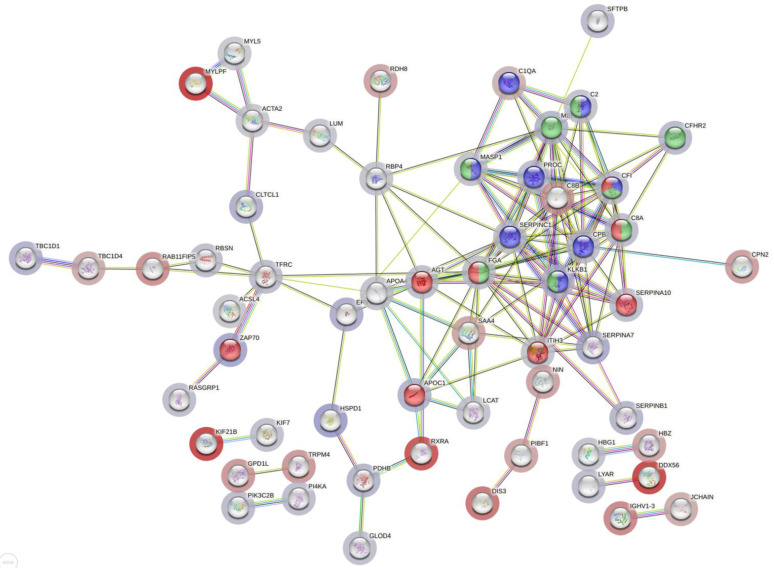
STRING PPI showing the protein interaction of common upregulated and downregulated proteins in moderate and/or under medication compared to recovered subjects. Edges represent protein–protein associations. Purple and light blue edges represent known interactions from experiments and curated databases, respectively. Red bubbles represent the network map of SARS-CoV-2 signaling pathways. Blue represents complement and coagulation cascades. Green represents the complement system.

**Figure 9 diagnostics-14-02533-f009:**
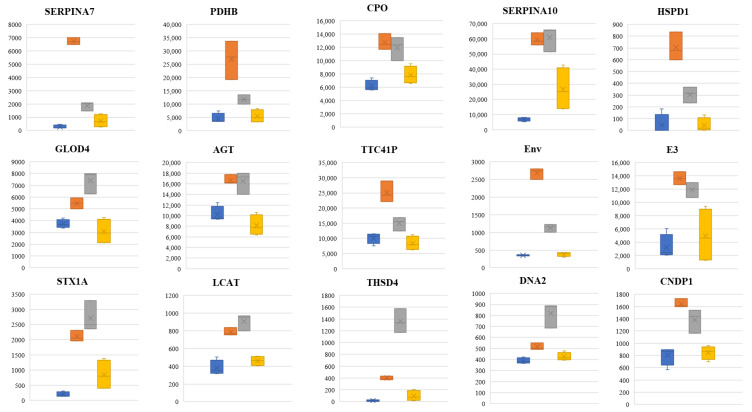
Box plot showing upregulated potential protein biomarkers. Blue represents healthy, orange represents moderates, gray represents under medication, and yellow represents recovered.

**Figure 10 diagnostics-14-02533-f010:**
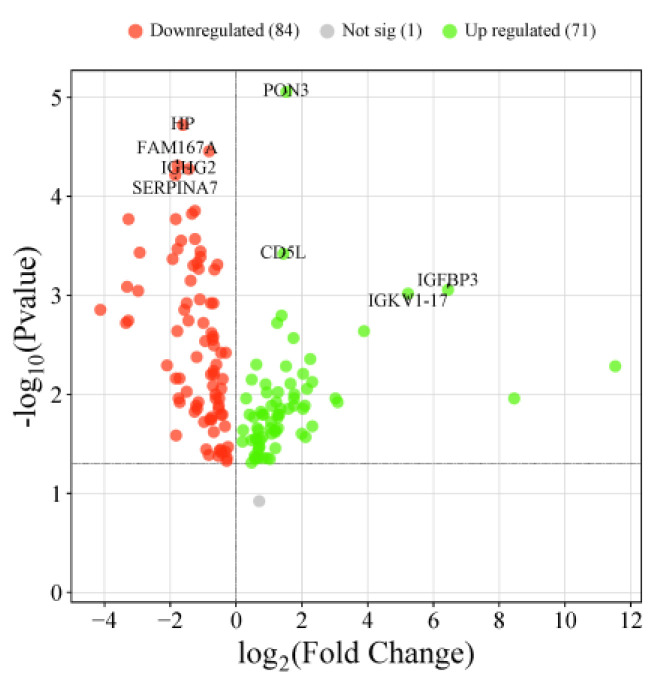
Volcano plot of upregulated and downregulated proteins in under medication patients compared to moderates.

**Table 1 diagnostics-14-02533-t001:** Demographic characteristics of studied subjects.

Serial ID	Category	Age	Nationality	Gender
8	Healthy	21	Saudi	Female
36	Healthy	21	Saudi	Male
55	Healthy	24	Saudi	Male
45	Healthy	27	Saudi	Male
25	Healthy	28	Saudi	Male
43	Healthy	38	Saudi	Female
59	Healthy	44	Saudi	Male
76	Healthy	44	Saudi	Male
56	Healthy	50	Saudi	Female
20	Healthy	57	Saudi	Female
42	Healthy	59	Saudi	Female
44	Healthy	63	Syrian	Male
7	Healthy	67	Syrian	Male
35	Healthy	28	Yemeni	Male
21	Healthy	34	Yemeni	Male
54	Healthy	28	Indian	Female
58	Healthy	68	Sudanese	Male
67	Healthy	44	Indonesian	Male
6	Healthy	23	Non-Saudi	Male
31	Healthy	22	Non-Saudi	Male
77	Moderate	42	Saudi	Female
72	Moderate	50	Saudi	Male
11	Moderate	58	Saudi	Male
5	Moderate	60	Saudi	Male
33	Moderate	64	Saudi	Female
2	Moderate	69	Saudi	Female
50	Moderate	38	Syrian	Male
23	Moderate	53	Syrian	Male
74	Moderate	50	Egyptian	Male
41	Moderate	62	Afghan	Male
53	Moderate	55	Yemeni	Male
29	Moderate	70	Yemeni	Male
66	Moderate	51	Bangladesh	Male
65	Moderate	56	Bangladesh	Female
75	Moderate	56	Sudanese	Male
69	Moderate	69	Canadian	Male
78	Moderate	36	Palestinian	Female
24	Moderate	37	Nepalese	Female
19	Moderate	38	Pakistani	Male
73	Moderate	41	Filipino	Male
32	Under MEDS	55	Saudi	Female
80	Under MEDS	56	Saudi	Male
14	Under MEDS	61	Saudi	Female
39	Under MEDS	67	Saudi	Male
48	Under MEDS	67	Saudi	Male
71	Under MEDS	73	Saudi	Male
26	Under MEDS	75	Saudi	Female
18	Under MEDS	NA	NA	NA
62	Under MEDS	29	Egyptian	Male
57	Under MEDS	39	Egyptian	Male
22	Under MEDS	40	Egyptian	Female
64	Under MEDS	54	Egyptian	Male
27	Under MEDS	48	Indian	Female
46	Under MEDS	66	Indian	Female
60	Under MEDS	43	Pakistani	Male
17	Under MEDS	45	Pakistani	Male
70	Under MEDS	62	Pakistani	Male
63	Under MEDS	68	Palestinian	Female
38	Under MEDS	51	Filipino	Male
13	Under MEDS	46	Syrian	Male
12	Recovered	47	Saudi	Female
16	Recovered	60	Saudi	Female
15	Recovered	72	Saudi	Male
49	Recovered	72	Saudi	Male
68	Recovered	86	Saudi	Female
34	Recovered	91	Saudi	Male
51	Recovered	41	Bangladesh	Female
4	Recovered	44	Bangladesh	Male
10	Recovered	60	Bangladesh	Male
61	Recovered	33	Sudanese	Male
40	Recovered	58	Sudanese	Male
37	Recovered	62	Sudanese	Female
3	Recovered	39	Egyptian	Male
1	Recovered	41	Moroccan	Female
28	Recovered	50	Filipino	Male
79	Recovered	56	Indian	Male
30	Recovered	61	Egyptian	Male
52	Recovered	61	Pakistani	Female
9	Recovered	65	Jordanian	Female
47	recovered	60	Sudanese	Male

Under MEDS = under medication, NA = not available.

## Data Availability

The original contributions presented in this study are included in the article/Appendix A. Further inquiries can be directed to the corresponding author.

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
