# Peer review of "Proteomic Profiling of COVID-19 Patients Sera: Differential Expression with Varying Disease Stage and Potential Biomarkers"

_diagnostics, 2024, doi:10.3390/diagnostics14222533_

Round 1

Reviewer 1 Report

Comments and Suggestions for Authors

The paper provides valuable insight into plasma proteome characteristics in COVID-19 patients and should be accepted after minor revision.

I find it important that the authors provide detailed data on the fold changes, variation and missing values for all the proteins that they managed to quantify. The magnitude of differences they observed should be considered and discussed in the text.

Another necessary correction should be done in every sentence, where the differentially expressed proteins that the authors observed are considered as biomarkers (lines 20 259 278 295 413). These can not be termed as "biomarkers" based on the data provided, but they are valuable potential biomarkers. In order to call any value a biomarker, a patient-wise measurement and necessary statistical assessment should be performed besides the analysis of pooled samples as was done by the authors.

Another issue with the data analysis is around the angiotensinogen, angiotensin I-III molecules. Detailed sample preparation procedures and parameters and output of the quantifying software  on peptide level should be evaluated to justify the attribution of differences in peptide intensities to specific molecule.

The work will benefit significantly if the collection studied contains paired samples of the same patient during different stages of disease (especially if a follow-up sample of a sick patient is available in recovered group). The authors should consider analysis of such paired samples individually.

If such paired samples are available, it should be noted anyway due to possible statistical implications. If not, this does not reduce the importance of work.

In addition, brief grammar revision is needed (see some mistakes I found below).

83-84 didn’t started - didn’t start

169 one-hundred-sixty-one - one hundred sixty-one 

170 one-hundred-forty-one - one hundred forty-one

226 one-hundred-thirty-eight - one hundred thirty-eight

231 one-hundred-thirteen - one hundred thirteen

243 vein - Venn

244 vein - Venn

281 two-hundred-eighty-eight - two hundred eighty-eight 

282  one-hundred-fifty-six -  one hundred fifty-six

313 mass spectrophotometry - mass spectrometry

Author Response

Reviewer 1

The paper provides valuable insight into plasma proteome characteristics in COVID-19 patients and should be accepted after minor revision.

  • I find it important that the authors provide detailed data on the fold changes, variation and missing values for all the proteins that they managed to quantify. The magnitude of differences they observed should be considered and discussed in the text.
    • Authors response: We thank the reviewer for his comment. A table showing the full details of the 288 differentially expressed proteins among all four categories in this study has been added in the supplementary material and cited in the main text Line 171. Also a discussion part regarding this data was added in Line 329- 338. Also kindly note that figure 10 was rectified, because during revision we have found that four proteins were mistakenly not included in the volcano plot.
  • Another necessary correction should be done in every sentence, where the differentially expressed proteins that the authors observed are considered as biomarkers (lines 20 259 278 295 413). These cannot be termed as "biomarkers" based on the data provided, but they are valuable potential  In order to call any value a biomarker, a patient-wise measurement and necessary statistical assessment should be performed besides the analysis of pooled samples as was done by the authors.
    • Authors response: we thank the reviewer for his comment and we have made the changes to reflect potential biomarkers throughout the manuscript as suggested.
  • Another issue with the data analysis is around the angiotensinogen, angiotensin I-III molecules. Detailed sample preparation procedures and parameters and output of the quantifying software on peptide level should be evaluated to justify the attribution of differences in peptide intensities to specific molecule.
    • Authors response: The detection limit of MS instruments is calibrated to measure relative number of low abundant proteins even in the mist of several highly abound molecules. In addition, we recognized the limit of any of the analytical instrument, hence we have focused on the coverage of identified proteins in this study. The Synapt G2 HDMS that was used in this study is very reliable to capture molecules within the detection limit of approximately 25fmol reliably. Therefore, the identified proteins that fulfilled the criteria were considered reliable. A clarification statement has been included in the revised manuscript Line 143-146.
  • The work will benefit significantly if the collection studied contains paired samples of the same patient during different stages of disease (especially if a follow-up sample of a sick patient is available in recovered group). The authors should consider analysis of such paired samples individually. If such paired samples are available, it should be noted anyway due to possible statistical implications. If not, this does not reduce the importance of work.
    • Authors response: We value reviewer’s suggestion. Unfortunately, the logistics during the pandemic did not allow us to collect samples from the same patient at multiple times. We truly agree with the reviewer, but for the reason given above, having the sample individual’s samples collected as cases and control during disease progression and recovery could have added strength to the study.
  • In addition, brief grammar revision is needed (see some mistakes I found below).

83-84 didn’t started - didn’t start

169 one-hundred-sixty-one - one hundred sixty-one 

170 one-hundred-forty-one - one hundred forty-one

226 one-hundred-thirty-eight - one hundred thirty-eight

231 one-hundred-thirteen - one hundred thirteen

243 vein - Venn

244 vein - Venn

281 two-hundred-eighty-eight - two hundred eighty-eight 

282  one-hundred-fifty-six -  one hundred fifty-six

313 mass spectrophotometry - mass spectrometry

  • Authors response: We apologize for the grammar and typo errors some of which were caused by too much reliability on auto corrections. All the suggested changes have been made in the revised manuscript.

Reviewer 2 Report

Comments and Suggestions for Authors

This study reported the proteomic profiling of COVID-19 patients sera and try to iendified differentially expressed proteins with  different disease stage as potential biomarkers. The study suggests the possible association of specific proteins with the SARS-COV-2 pathogenesis and their potential use as disease biomarkers and drug targets. This has some important implications for the diagnosis and treatment of COVID-19.  However, the study can be improved with the following modifications.

1. In line 18,"in addition to the lack of approved anti-viral drug", it should be revised as there are already some drugs for COVID-19.

2. In lines 79-80, These included 20 patients (<15 years old) from each of the following categories. Here for "<15 years old", there may be something wrong. In the result, they said that the ages ranged from twenty-one to nighty-one years old.

3. in lines 156-157, Owing to low throughput of LC-MS- proteomics analysis, 20 samples in each cohort  category, were pooled together and run for the assessment of proteomic profiles. Please explain in detail. Why a total of 80 samples changed to 14 samples in Figure 2.

4. Figure 1 shows that P >0.05 in two places and P <0.05 in one place, which one is correct?

5. Bioinformatics analysis should be in-depth and the results should be processed with some degree of condensation before being presented.

6. Importantly, the author should learn and compare with the paper published in 2020. (Shen B, Yi X, Sun Y, et al. Proteomic and metabolomic characterization of COVID-19 patient sera[J]. Cell, 2020, 182(1): 59-72. e15.). Then the author should present some different and new results.

Author Response

Reviewer 2

This study reported the proteomic profiling of COVID-19 patients sera and try to identified differentially expressed proteins with different disease stage as potential biomarkers. The study suggests the possible association of specific proteins with the SARS-COV-2 pathogenesis and their potential use as disease biomarkers and drug targets. This has some important implications for the diagnosis and treatment of COVID-19.  However, the study can be improved with the following modifications.

  1. In line 18,"in addition to the lack of approved anti-viral drug", it should be revised as there are already some drugs for COVID-19.
    • Authors response: We thank the reviewer for this comment. This statement has been amended and removed from the abstract and also from the main text.
  1. In lines 79-80, These included 20 patients (<15 years old) from each of the following categories. Here for "<15 years old", there may be something wrong. In the result, they said that the ages ranged from twenty-one to nighty-one years old.
    • Authors response: We apologize for the type error. All patients are > 15 years old. This has now been corrected in the revised manuscript Line 78.
  1. in lines 156-157, Owing to low throughput of LC-MS- proteomics analysis, 20 samples in each cohort category, were pooled together and run for the assessment of proteomic profiles. Please explain in detail. Why a total of 80 samples changed to 14 samples in Figure 2.
    • Authors response: We have described under materials and methods section that each sample compositions of an analysis group were pooled into a cohort prior to MS analysis. Therefore, each cohort pooled samples subjected to multiple LC/MS runs (3 to 4 times) and average of the multiple runs were evaluated for statistical analysis. We recognized the inherent limitation of analysis of pooled samples in any analysis platforms including MS-based analysis with potential risks of confounding the results if proper care is not as widely reportedly. However, in this study we have taken steps, to mitigate potential artifacts arising from pooling samples. We always undertake initial screening step in maintaining homogeneity within an analysis group prior to pooling. As this is a discovery phase of the study, therefore, all individual samples in each of the pools together with other samples will always be used in the validation for individual variability of the observed changes in the pooled samples. We are cognizance of this and care is always taken that sample representation is prerequisite to achieving quality and reliable data. This aspect has been incorporated in the revised manuscript Line 164-168.

  1. Figure 1 shows that P >0.05 in two places and P <0.05 in one place, which one is correct?
    • Authors response: we thank the reviewer for his comment. Actually, both P values are correct. As mentioned in the discussion Lines 428-430, there were an “age semi matching” between controls and infected groups (P values < 0.05). However, for the other comparisons, no significant statistical difference was observed in age (P > 0.05).
  1. Bioinformatics analysis should be in-depth and the results should be processed with some degree of condensation before being presented.
    • Authors response: We thank the reviewer for this important concern. We have taken adequate measures into consideration in our data analysis. We have applied stringent false discovery rate (FDR) measures available in the Progenesis analysis program that we have used. In addition, we have considered only proteins with more than 1.5-fold change and p value of <0,05 as statistically significant. Furthermore, we have avoided multiple testing of observed changes, and incorporated adjusted p-value or q-value number of peptides used in the identification as well as applied power threshold in the filter in the selection of our potential differentially expressed protein biomarkers as detailed under 2.4 section.
  1. Importantly, the author should learn and compare with the paper published in 2020. (Shen B, Yi X, Sun Y, et al. Proteomic and metabolomic characterization of COVID-19 patient sera[J]. Cell, 2020, 182(1): 59-72. e15.). Then the author should present some different and new results.
    • Authors response: we thank the reviewer for his comment. A comparison with the findings of this paper was amended as per the reviewer comment Line 338-342 and in Lines 377-381.

Round 2

Reviewer 2 Report

Comments and Suggestions for Authors

The authors have responded well to my comments and have revised the paper accordingly.